# SARS-CoV-2 RNA Extraction Using Magnetic Beads for Rapid Large-Scale Testing by RT-qPCR and RT-LAMP

**DOI:** 10.3390/v12080863

**Published:** 2020-08-07

**Authors:** Steffen Klein, Thorsten G. Müller, Dina Khalid, Vera Sonntag-Buck, Anke-Mareil Heuser, Bärbel Glass, Matthias Meurer, Ivonne Morales, Angelika Schillak, Andrew Freistaedter, Ina Ambiel, Sophie L. Winter, Liv Zimmermann, Tamara Naumoska, Felix Bubeck, Daniel Kirrmaier, Stephanie Ullrich, Isabel Barreto Miranda, Simon Anders, Dirk Grimm, Paul Schnitzler, Michael Knop, Hans-Georg Kräusslich, Viet Loan Dao Thi, Kathleen Börner, Petr Chlanda

**Affiliations:** 1Center of Infectious Diseases, Virology, Heidelberg University Hospital, 69120 Heidelberg, Germany; steffen.klein@bioquant.uni-heidelberg.de (S.K.); thorsten.mueller@uni-heidelberg.de (T.G.M.); Dina.Khalid@med.uni-heidelberg.de (D.K.); Vera.Sonntag-Buck@med.uni-heidelberg.de (V.S.-B.); Anke-Mareil.Heuser@med.uni-heidelberg.de (A.-M.H.); Baerbel.Glass@med.uni-heidelberg.de (B.G.); angelika.schillak@med.uni-heidelberg.de (A.S.); Andrew.Freistaedter@med.uni-heidelberg.de (A.F.); sophie.winter@bioquant.uni-heidelberg.de (S.L.W.); Liv.Zimmermann@med.uni-heidelberg.de (L.Z.); Tamara.Naumoska@med.uni-heidelberg.de (T.N.); f.bubeck@stud.uni-heidelberg.de (F.B.); Stephanie.Ullrich@med.uni-heidelberg.de (S.U.); Isabel.BarretoMiranda@med.uni-heidelberg.de (I.B.M.); dirk.grimm@bioquant.uni-heidelberg.de (D.G.); paul.schnitzler@med.uni-heidelberg.de (P.S.); hans-georg.kraeusslich@med.uni-heidelberg.de (H.-G.K.); VietLoan.DaoThi@med.uni-heidelberg.de (V.L.D.T.); 2Schaller Research Groups, Center of Infectious Diseases, Virology, Heidelberg University Hospital, 69120 Heidelberg, Germany; 3Center for Molecular Biology of Heidelberg University (ZMBH), 69120 Heidelberg, Germany; m.meurer@zmbh.uni-heidelberg.de (M.M.); d.kirrmaier@zmbh.uni-heidelberg.de (D.K.); s.anders@zmbh.uni-heidelberg.de (S.A.); m.knop@zmbh.uni-heidelberg.de (M.K.); 4German Cancer Research Center (DKFZ), 69120 Heidelberg, Germany; 5Center of Infectious Diseases, Clinical Tropical Medicine, Heidelberg University Hospital, 69120 Heidelberg, Germany; ivonne.morales@uni-heidelberg.de; 6Center of Infectious Diseases, Integrative Virology, Heidelberg University Hospital, 69120 Heidelberg, Germany; ina.ambiel@med.uni-heidelberg.de; 7German Center for Infection Research (DZIF), 69120 Heidelberg, Germany; 8DKFZ-ZMBH Alliance, 69120 Heidelberg, Germany

**Keywords:** coronavirus, SARS-CoV-2, COVID-19, diagnostics, RT-qPCR, RT-LAMP, magnetic bead RNA purification, RNA virus, pandemic, high-throughput screening

## Abstract

Rapid large-scale testing is essential for controlling the ongoing pandemic of severe acute respiratory syndrome coronavirus 2 (SARS-CoV-2). The standard diagnostic pipeline for testing SARS-CoV-2 presence in patients with an ongoing infection is predominantly based on pharyngeal swabs, from which the viral RNA is extracted using commercial kits, followed by reverse transcription and quantitative PCR detection. As a result of the large demand for testing, commercial RNA extraction kits may be limited and, alternatively, non-commercial protocols are needed. Here, we provide a magnetic bead RNA extraction protocol that is predominantly based on in-house made reagents and is performed in 96-well plates supporting large-scale testing. Magnetic bead RNA extraction was benchmarked against the commercial QIAcube extraction platform. Comparable viral RNA detection sensitivity and specificity were obtained by fluorescent and colorimetric reverse transcription loop-mediated isothermal amplification (RT-LAMP) using a primer set targeting the N gene, as well as RT-qPCR using a primer set targeting the E gene, showing that the RNA extraction protocol presented here can be combined with a variety of detection methods at high throughput. Importantly, the presented diagnostic workflow can be quickly set up in a laboratory without access to an automated pipetting robot.

## 1. Introduction

Severe acute respiratory syndrome coronavirus 2 (SARS-CoV-2), the causative agent of coronavirus disease 2019 (COVID-19), was first described in the city of Wuhan in China in December 2019 and spread globally thereafter causing pandemic. To slow its spread, large-scale diagnostics and the enforcement of strict public health measures were implemented in many countries. The current standard test for SARS-CoV-2 detection and diagnosis is based on viral RNA extraction from a pharyngeal swab followed by highly sensitive reverse transcription and quantitative PCR (RT-qPCR). Several primer sets targeting one or more of the SARS-CoV-2 genes—nucleocapsid (N), envelope protein (E), S glycoprotein (S), or RNA-dependent RNA polymerase (RdRp)—have been used [1]. A two-step testing procedure using primer sets targeting the E gene for initial screening followed by the RdRp gene to confirm positive samples is recommended by the German Consiliary Laboratory for Coronaviruses [2]. The unprecedented global demand for commercial RNA extraction kits and ensuing shortage of these reagents [3] led to the establishment of several diagnostic workflows performed on patient samples with or without an intermediate RNA extraction step [4,5,6,7]. Viral RNA isolation from clinical samples depends on the rapid inactivation of viral particles, typically by detergent solubilization, and on the denaturation of omnipresent RNases [8]. The latter may be accomplished by the use of chaotropic chemicals, such as guanidinium salts [9] or non-specific proteases that are active on both native and denatured proteins, such as proteinase K. In either case, after virus particle lysis, RNA must be purified, since guanidinium salts, proteinase K and organic solvents inhibit the subsequent RT-qPCR step. RNA can be separated from proteins either by liquid phase separation using chloroform-aqueous emulsions after lysis with commercially available Trizol (a mixture of guanidinium thiocyanate and acid phenol) or by means of solid-phase separation using silica [10]. Nucleic acid-binding to negatively charged silica (SiO_2_) is facilitated by guanidinium salts and the basic pH of the lysis buffer [11]. To achieve a high nucleic acid binding capacity, silica-based nucleic acid extraction methods use either porous silica matrices that are embedded in a column (spin column) [12], a tip (TruTips) [13], or a suspension of microparticles. Microparticles can be separated from the lysate either by centrifugation or by a magnetic field provided that the microparticles’ dense iron-containing cores are coated with porous silica [14].

The protocol established in this study aimed at extracting SARS-CoV-2 RNA from respiratory patients’ swabs (oropharyngeal and nasopharyngeal) and is based on the magnetic bead-based nucleic acid extraction protocol that was published by He et al. [15] and on the protocol from Chemicell GmbH, who provided the SiMAG-N-DNA magnetic beads. We opted for silica magnetic beads because of their relatively easy manufacturing and sustainable availability, and because specialized plasticware (spin columns, modified tips) is not required to perform their separation. Magnetic bead RNA extraction was performed in 96-well plates in combination with a magnet plate optimized for 96 deep-well plates. The portable manual pipetting system Liquidator 96, which is less expensive than automated pipetting robots, was used to minimize the pipetting and handling errors (Figure 1A). Here, we show that the magnetic bead-based protocol yields RNA extracts comparable to the commercially available QIAcube viral RNA extraction kit, as determined by the commonly applied detection methods RT-qPCR and reverse transcription loop-mediated isothermal amplification (RT-LAMP) [16].

## 2. Materials and Methods

### 2.1. Clinical Samples and Sample Lysis

A selection of upper respiratory tract specimens (flocked swabs in Amies medium, eSwab Copan, Brescia, Italy) sent to the diagnostics laboratory of the Heidelberg University Hospital between April and May 2020 for SARS-CoV-2 PCR were used for the study. Surplus material from a total of 77 swab samples were collected from 17 SARS-CoV-2 positive and 60 negative patients, which were used for SARS-CoV-2 testing by RT-qPCR. The swab samples were either used at the day of collection (two positive, 20 negative) or were frozen, stored at −20 °C, and thawed just before the RNA extraction (15 positive, 40 negative). Positive and negative samples were further diluted and used in replica to generate 88 positive and 76 negative samples in order to facilitate replicate testing.

The following steps were performed in a biosafety level 2 (BSL-2) laboratory according to standard microbiological and diagnostic practices. To extract SARS-CoV-2 RNA from pharyngeal swabs, a lysis buffer containing 5 M guanidinium thiocyanate, 40 mM dithiothreitol, 20 µg/mL glycogen, 1% Triton X-100, and buffered with 25 mM sodium citrate to pH 8 was used. Internal control (IC) for RT-qPCR (5 µL/sample) (Tib-Molbiol, Berlin, Germany) was added into the lysis buffer just before use. 140 µL lysis buffer and 140 µl sample were vigorously vortexed in a 1.5 mL Eppendorf tube for 10 sec and incubated for 10 min at room temperature inside a BSL-2 laminar flow cabinet to ensure both rapid virus deactivation and RNase denaturation. Lysates (280 µL) were transferred into a 96 deep-well plate (Greiner AG, Kremsmünster, Austria) with a maximum working volume of 500 µl per well, which is compatible with the Liquidator 96 pipetting system.

### 2.2. RNA Extraction Using Magnetic Beads in a 96-Well Plate Format

Just before the RNA extraction, SiMAG-N-DNA magnetic beads (Chemicell, Berlin, Germany) were washed three times in RNase-free water. The aqueous magnetic bead stock solution of 100 µg/µl was added into absolute ethanol in order to obtain a working solution of 5 µg/µL. Using a multichannel pipette, 200 µL of magnetic bead solution was transferred into the 96 deep-well plate containing the pharyngeal sample lysates. To facilitate the adsorption of the nucleic acid onto the magnetic beads, the 96-deep-well plate was placed on an orbital shaker MS3 (IKA, Staufen, Germany) for 8 min at 500–1000 rpm, before the mixture was resuspended (10×) using a Liquidator 96, Model 200 µl (Mettler Toledo, Columbus, OH, USA) (Figure 1A,B) and placed again on the orbital shaker for additional 7 min. Subsequently, the plate was placed on a magnet plate for 96-deep-well plates (Magtivio, Nuth, The Netherlands) (Figure 1C) for 10 min to allow the magnetic beads to form rings on the bottom of the wells (Figure 1D,E). The 96-well plate was visually inspected to ensure that all of the pellets were formed. The clear supernatant was discarded using the Liquidator 96 and magnetic beads were washed three times with 200 µL 70% ethanol. For each washing step, the 96-well plate was removed from the magnet and pellets were resuspended (10×) using a Liquidator 96. The plate was placed back on the magnet for 1 min until the beads formed visible rings. After three washing steps, the magnetic beads were briefly rinsed with 60 µL RNase-free water in order to remove any residual ethanol while the plate was kept on the magnet. The 96-well plate was visually inspected to ensure that none of the pellets was removed. Finally, nucleic acids adsorbed onto the surface of the magnetic beads were eluted: 50 µL RNAse-free water was added to each well, the 96-well plate was removed from the magnet, resuspended (10×), and vortexed for 5–10 min. The 96-well plate was placed back on the magnet until rings of magnetic beads were formed and 50 µL eluate was transferred to a new 96-well PCR plate. A detailed step-by-step procedure and a complete list of all materials and instruments used for this magnetic bead RNA extraction protocol can be found in the Appendix A.

### 2.3. RNA Extraction Using QIAcube

The QIAamp Viral RNA body fluid kit was carried out with manual lysis according to the manufacturer’s protocol to compare the performance of the magnetic bead RNA extraction (Qiagen, Hilden, Germany). The sample input volume was 140 µl, the volume of IC per sample was 10 µL, and the elution volume was set to 100 µL.

### 2.4. SARS-CoV-2 RNA Detection by RT-qPCR

For RT-qPCR detection of the SARS-CoV-2 envelope protein gene (E gene), we adopted a widely used protocol based on Corman et al. [2]. For the Mastermix, 0.5 µl of Primer/Probe LightMix**^®^** Modular SARS and Wuhan CoV E-gene (Tib-Molbiol, Berlin, Germany) and 0.5 µl of LightMix**^®^** Modular EAV RNA Extraction Control (Tib-Molbiol, Berlin, Germany) was mixed with 4.9 µl RNase-free water, 4 µl LightCycler**^®^** Multiplex RNA Virus Master (Roche, Basel, Switzerland), and 0.1 µl Reverse Transcriptase Enzyme (supplied with LightCycler**^®^** Multiplex RNA Virus Master kit, Berlin, Germany) per sample. The following primers and probe were used (provided by Tib-Molbiol, Germany): fwd 5**′**-ACAGGTACGTTAATAGTTAATAGCGT-3**′**, rev 5**′**-ATATTGCAGCAGTACGCACACA-3**′**, probe FAM-ACACTAGCCATCCTTACTGCGCTTCG-BBQ; These primers are specific for a 113 bp amplicon from position 26141–26253 of the SARS-CoV-2 genome (GenBank NC_004718). 10 µl of Mastermix was distributed into a 96-well PCR plate and 10 µl of purified RNA from patient samples was added using the Liquidator 96, model 10 µl. RT-qPCR was performed using a LightCycler**^®^** 480 Instrument II (Roche, Basel, Switzerland) with 5 min of reverse transcription at 55 °C, initial denaturation at 95 °C for 5 min, and subsequent 45 amplification cycles with 95 °C for 5 sec, 60 °C for 15 sec, 72 °C for 15 sec, and finally cooling to 40 °C for 30 sec. Cycle threshold (CT) was determined, where the fluorescence signal of the amplification reaction was above the background fluorescence using the LightCycler software (Roche). Data analysis on raw CTs was performed in Excel and GraphPad Prism (GraphPad Software, San Diego, CA, USA), and 95% “exact” Clopper-Pearson confidence intervals were calculated using MedCalc (MedCalc Software, Ostend, Belgium).

### 2.5. Estimation of the Detection Limit of Magnetic Bead RNA Extraction Using MS2 RNA Spike-In

Five µl diluted MS2 RNA (Roche, Basel, Switzerland) containing 6.1 × 10^1^ to 6.1 × 10^6^ molecules was spiked into 140 µl lysis buffer before a 140 µl patient sample was added. Magnetic bead RNA extraction was performed as described in Section 2.2. RT-qPCR for MS2 was performed with a one-step RT-qPCR reaction (Tib-Molbiol, Berlin, Germany) using the following primers and probe: fwd 5′-GAGTGTTTACAGTTCCGAA-3′, rev 5′-CCCCTTTCTGGAGGTACATATTCATA-3′, and probe Cy5-AATAGATCGGGCTGCCTGTAAGGAGC-BBQ. 10 µl of RNA was used per RT-qPCR, covering a range from 10^1^ to 10^6^ MS2 RNA molecules per reaction. RT-qPCR cycling program for MS2 RNA amplification was performed as described in Section 2.4.

### 2.6. SARS-CoV-2 RNA Detection by Colorimetric RT-LAMP

RT-LAMP detection of the SARS-CoV-2 N gene was based on the protocol by Dao Thi et al. [7]. The RT-LAMP primer set used in this study was targeted against the SARS-CoV-2 N gene [17]. The sequences and concentrations of all oligonucleotides in the 10x primer mix used for the RT-LAMP assays can be found in our recent publication [7]. For the colorimetric LAMP assay, the master mix consisted of 6.25 µl WarmStart^®^ Colorimetric LAMP 2X Master Mix M1800 (New England Biolabs, Ipswich, MA, USA) and 1.25 µl 10× LAMP primer mix targeting the N gene [17] per reaction. Immediately afterwards, 7.5 µl of the freshly prepared reaction mix was distributed into 96-well plates and 5 µl of purified RNA was added using a Liquidator 96, Model 20 µl. The plate was sealed with a transparent adhesive foil and subsequently incubated at 65 °C for 30 min in a 96-well-PCR block with heated lid (75 °C). Absorbance was measured at 434 nm and 560 nm wavelengths in a Spark^®^ Cyto or Infinite M200 plate reader (Tecan, Männedorf, Switzerland). Phenol red absorbance spectra change in response to the acidification of the reaction (the absorbance at 434 nm wavelength is increased and the absorbance at 560 nm wavelength is decreased). To measure pH changes during the reaction, the difference between optical densities was calculated (ΔOD = OD_434nm_ − OD_560nm_) and samples with ΔOD < 0.3 were classified as negative.

### 2.7. SARS-CoV-2 RNA Detection by Fluorescent RT-LAMP

For the fluorescent LAMP assay, the same primer set as described in 2.6 was used. The master mix consisted of 6.25 µl WarmStart^®^ LAMP Kit 2X Master Mix E1700 (New England Biolabs, Ipswich, MA, USA), 1.25 µl 10× LAMP primer mix targeting the N gene [17] and 0.25 µl of 50× fluorescent dye (Syto-9, supplied with the RT-LAMP kit) per reaction. 7.75 µl of the mix was distributed in a 96-well plate and 4.75 µl purified RNA was added before the plate was sealed and incubated at a constant temperature of 65 °C using a LightCycler^®^ 480 Instrument II (Roche, Basel, Switzerland). Real-time fluorescence was detected with intervals of 1 min for a duration of 90 min. Time of amplification (TOA) was determined based on a fluorescence threshold, where the fluorescence signal of the amplification reaction was above the background fluorescence, using the LightCycler software. Samples with TOA > 25 min were classified as negative.

### 2.8. Ethics Statement

The presented work was done with the intention to support SARS-CoV-2 diagnostics and improve emergency preparedness and response to COVID-19. Pseudo-anonymized surplus material from samples that had been collected for SARS-CoV-2 testing were used to establish and validate the protocol presented here. This work complies with the German Act concerning the Ethical Review of Research Involving Humans, permitting use of patient samples collected to perform the testing in question, for development and improvement of diagnostic assays.

## 3. Results

The magnetic bead RNA extraction protocol was established in a 96-well plate format as part of the detection workflow (Figure 2). The complete workflow from RNA extraction to RNA detection can be conducted in less than 4–5 h.

### 3.1. RNA Extraction Using Magnetic Beads Yields RT-qPCR Results That Are Comparable to a Commercial Extraction Kit

We first applied both protocols on one SARS-CoV-2 positive pharyngeal swab sample, which was diluted in RNase-free water prior to RNA isolation in a 10-fold dilution series up to 10^5^ fold, in order to compare the magnetic bead RNA extraction protocol with the commercial QIAcube extraction method. The extracted RNA was subjected to RT-qPCR using E gene primers. Viral RNA was detected in samples diluted up to 10^5^ fold after either QIAcube or magnetic bead RNA extraction (Figure 3A,C). RT-qPCR of the extracted RNA by either of the two methods showed approximately equidistant amplification curves with an interval of 4 CT values for each 10-fold dilution step (Figure 3B,D).

A dilution series of RNA from MS2 bacteriophage was added to the swab sample prior to magnetic bead RNA extraction and subjected to RT-qPCR using MS2 primers to further evaluate linearity of the magnetic bead RNA extraction method across a broad range of defined RNA inputs (Figure 3E,F). CT values were linear over five orders of magnitude with a goodness of fit of R^2^ = 0.9804. The lower limit of detection (~CT 40) was less than 10 RNA copies/reaction (Figure 3E).

### 3.2. RT-qPCR Performed on Magnetic Bead Extracted RNA Shows High Detection Sensitivity and Specificity

In order to evaluate the magnetic bead RNA extraction protocol on larger sample sets, 88 SARS-CoV-2 positive and 76 negative samples were generated from 17 SARS-CoV-2 positive patients and from 60 persons negative for SARS-CoV-2, respectively. The samples were subjected to three independent magnetic bead RNA extractions as well as to QIAcube extractions. CT values of 82 positive samples obtained by RT-qPCR using primers for the E gene and IC values from either magnetic bead or QIAcube extraction were compared. In the remaining six samples, CT values for the E gene were not detected after magnetic bead RNA purification but detected CT values for IC confirmed successful RNA extraction. Noteworthy, all six samples were previously frozen and had a CT value higher than 35 as determined by RT-qPCR after the QIAcube RNA extraction. The E gene CT values that were obtained from the magnetic bead and QIAcube RNA extraction were in good agreement as shown by the slope of the linear regression of 1.037 (Figure 4A). The average IC CT of 29.3 (SD = 1.2, *n* = 82) after magnetic bead extraction was the same as compared to the average IC CT of 29.3 from the QIAcube RNA extractions (SD = 2.3, *n* = 82) (Figure 4B,C), showing that the magnetic bead RNA extraction protocol is robust. These results also illustrate that quickly washing the beads with water to remove residual ethanol (which could inhibit PCR) does not lead to a substantial loss of RNA. The SD of CT values for IC should be calculated to determine false negatives due to RNA loss and judge the quality of the RNA extraction. The SD of the CT values for IC should be smaller than 3 CT values and samples with IC CT values with 2× SD lower IC CT values than the average IC CT should be repeated.

Because assays performed in a 96-well plate format can be prone to cross-contaminations, we evaluated the sensitivity and specificity of the three independent magnetic bead RNA extractions. The samples were distributed in small groups of positive and negative samples on the 96-well plate. After magnetic beads RNA extraction and RT-qPCR or RT-LAMP analysis the results were compared to the results of the RT-qPCR after QIAcube extraction to determine true and false positives (TP and FP) and true and false negatives (TN and FN). For the magnetic beads extracted samples, a CT cutoff of 40 was used to determine if a sample is defined as positive. The sensitivity was calculated (TP/(TP+FN)) for subsets of samples in specific CT ranges (0–25, 25–30, 30–35, 35–40) (Figure 4D). For all subsets up to CT 35, no false negatives were detected, which leads to a sensitivity of 100% with a 95% confidence interval (CI) of 86–100%. In the range between CT 35–40, six false negatives were counted leading to a sensitivity of 54% (CI = 25–81%). These results indicate that false negatives were samples with high CT values and the negative results might be due to fluctuation of the viral load around the limit of detection. In total, three false positives were detected, resulting in a specificity of 96% (CI = 89–99%) (Figure 4E).

### 3.3. RNA Extraction Using Magnetic Beads is Compatible with the RT-LAMP Assay

We also recently explored the RT-LAMP assay as a valid alternative RNA detection method due to shortages in RT-qPCR reagents [7]. We compared the standard pipeline QIAcube RNA extraction followed by RT-qPCR detection of the E gene with magnetic RNA extraction followed by RT-LAMP detection using primer sets targeting the N gene, with either colorimetric (Figure 5A–C, Appendix A) or fluorescent (Figure 5D–F, Appendix A) read-out. In total, we tested RNA extracted from 52 positive and 29 negative swab samples.

During DNA synthesis, the formation of a phosphodiester bond results in the release of a molecule of pyrophosphate and a proton causing a gradual acidification of the reaction mix. The detection principle of colorimetric RT-LAMP is based on monitoring pH changes during the DNA amplification, which only occurs in positive samples. Typically, phenol red, which changes color from red to yellow when the pH is lowered, is used as a pH indicator [18]. The color change can be determined by measuring the difference between the wavelengths of the two absorbance maxima of phenol red. The majority of isolated RNA samples with a CT ≈ 30 value obtained from RT-qPCR using the E gene yielded a color change with ΔOD values between 0.3 and 0.4 as shown in Figure 5A. Therefore, we used a ΔOD value of 0.3 as a threshold for positive samples. The majority of samples with higher CT values, especially between CT 35 and CT 40, scored negative (ΔOD < 0.3), while all negative RT-qPCR samples also scored negative in the RT-LAMP assay. Using a CT cutoff of 30, the overall sensitivity of the colorimetric assay with magnetic bead-isolated RNA was 97% (95% CI = 83–100%) (Figure 5B), with a specificity of 100% (95% CI = 88–100%) (Figure 5C).

The fluorescent RT-LAMP assay is based on a fluorescent dye that intercalates into amplifying DNA strands, allowing for their detection in real time. Magnetic bead isolated samples with CT ≈ 30 led to a fluorescent signal at early time points (approx. 20–25 min) as compared to negative samples (>40 min). Therefore, we used the time point 25 min as a threshold for positive samples. Similar to the colorimetric read-out, the majority of samples with higher CT values, especially between CT 35 and CT 40, scored negative (> 40 min) (Figure 5D). All negative RT-qPCR samples also scored negative (>70 min). Using a CT cutoff of 30, the overall sensitivity of the fluorescent assay was 100% (95% CI = 88–100%) (Figure 5E), with a specificity of 100% (95% CI = 88–100%) (Figure 5F).

These results are well in agreement with the recently reported sensitivity cutoff at CT 30 for the RT-LAMP assay [7] and, thus, confirm that RNA purified by the presented magnetic bead extraction is compatible with RT-LAMP detection without lowering its sensitivity.

## 4. Discussion

Within the last decades, the frequency of emerging virus outbreaks has increased globally [19,20], possibly as a result of cumulative anthropogenic environmental changes that increase the risk of zoonotic transmission [21]. Due to globalization, many of the outbreaks have increased pandemic potential and pose a burden on society and health systems. The currently ongoing SARS-CoV-2 pandemic emphasizes the urgency of appropriate preparedness and response. Before a therapeutic or vaccine exists, the early identification and isolation of positive patients remains the most effective way to inhibit further human-to-human spread and mitigate the disease outbreak. In the case of a respiratory viral disease, such as influenza or COVID-19, pharyngeal swabs are collected and tested for the presence of viral RNA. RNA isolation prior to detection is a pivotal step to ensure high specificity and sensitivity of detection methods. To this end, robust and high-throughput nucleic acid isolation methods with high manufacturing and distribution capacity must be available. However, the majority of commercially available RNA purification kits are cost-ineffective and often rely on multiple components that are not easily replaceable, and their supply can often not be guaranteed. In addition, buffer compositions are not provided. Thus, overall, commercial kits do not offer enough flexibility and availability when it comes to a large epidemic or pandemic.

Here, we provide a magnetic bead-based RNA extraction protocol that is, to a large extent, producer independent, does not rely on unique components that are difficult to replace, and is scalable for mass testing. Because of the large surface binding capacity and rapid separation in solution by magnetic field, silica coated magnetic beads are compatible with a variety of plasticware and they can be used in a high throughput multiwell format [4] as well as in small scale testing [15]. In addition, magnetic beads can be prepared from widely available chemicals in a laboratory with basic equipment [22]. Recently, a magnetic bead RNA purification protocol has been established for automated, high throughput SARS-CoV-2 diagnostics by the COVID-19 Crick Consortia [4]. However, a sophisticated infrastructure is required to carry out the protocol, and the protocol is partially dependent on commercial buffers. Our aim was to provide a robust protocol that can be rapidly implemented and carried out in most of the laboratories around the world equipped with a magnetic plate, a multichannel pipette, or with a manual 96 pipetting device. We established a magnetic bead RNA isolation protocol that takes approximately 90 min using a single 96-well plate as part of a workflow for SARS-CoV-2 diagnostic. Although it is feasible to perform four extractions per day for one person, the rate-limiting and the most labor-intensive step of the presented workflow is the sample transfer from swab tubes to 96-well plates. Special care should be taken regarding sample handling prior to RNA isolation, and the samples should be kept at 4 °C and processed at the day of collection. A single freeze-thawing cycle of a SARS-CoV-2 positive swab sample that was 2× diluted results in decreased sensitivity of RT-qPCR by approximately 7 CT values (CT_frozen_- CT_fresh_ = 7.4; SD = 3.5, *n* = 19). In contrast to other magnetic bead based nucleic acid purification protocols [4,15] that rely on an air-drying step to remove residual ethanol before elution, we rinse magnetic beads with a small amount of RNase-free water. The introduction of this step made our protocol more robust, since air-drying in a 96-well plate is slow and uneven. Furthermore, air-drying did not lead to complete removal of ethanol, which interfered with the subsequent RT-qPCR even at low concentrations. We report a similar RNA extraction yield when compared to the commercial QIAcube extraction kit and validate that the quality of the RNA extract is suitable for RT-qPCR and RT-LAMP detection.

RT-LAMP relies on a different DNA polymerase than RT-qPCR and, thus, provides a supplier-independent alternative to RT-qPCR. Additionally, RT-LAMP is faster and cheaper when compared to RT-qPCR and it does not require a thermocycler [7]. As we have recently reported [7], RT-LAMP to detect SARS-CoV-2 RNA has a decreased sensitivity compared to RT-qPCR: RT-LAMP with the primer set against the N gene as used in this study failed to detect most of the samples with a CT value over 30 measured by RT-qPCR using the E-Sarbeco primers. Efforts are underway to increase the sensitivity of the RT-LAMP assay, but this is out of the scope of this work. Here, we showed that the presented magnetic bead extraction is compatible with RT-LAMP and does not lower its sensitivity when compared to a QIAcube purification method performed in our previous study [7]. Either RT-qPCR or RT-LAMP can be used as an independent detection method and both methods do not have to be run in parallel. Due to higher sensitivity, we recommend using RT-qPCR as a detection method after magnetic bead RNA extraction. If resources are limited or large-scale testing is needed, RT-LAMP offers an alternative detection approach; however, its lower sensitivity must be considered for now.

Although it is possible to perform RT-qPCR [23,24] and RT-LAMP on unpurified patient samples [7,25], the combination with our magnetic bead RNA extraction protocols increases the sensitivity. Due to the low cost, high-throughput compatibility, and independence from sophisticated laboratory equipment such as automated RNA extraction kits and RT-qPCR machines, the combination of magnetic bead RNA extraction and RT-LAMP offers a framework for diagnostic preparedness in the case of mass-scale testing.

In conclusion, we report a detailed step-by-step RNA extraction protocol from pharyngeal swabs, which is based on magnetic beads and does not depend on commercial extraction kits and reagents. Our protocol was validated in a 96-well plate format by the detection of SARS-CoV-2 RNA by RT-qPCR and RT-LAMP and provides reliable RNA extraction for a diagnostic pipeline that can be rapidly deployed during the SARS-CoV-2 pandemic and is also accessible to regions with insufficient laboratory capacities. This might be of importance in case of possible future shortages of commercial RNA extraction kits and enables diagnostic laboratories to be well prepared in case of a new rise of SARS-CoV-2 cases. Our protocol can be easily adapted to fully automated liquid handling robotic systems and is very likely suitable for isolation and downstream detection assays for any kind of RNA virus isolated from pharyngeal swabs.

## Figures and Tables

**Figure 1 viruses-12-00863-f001:**
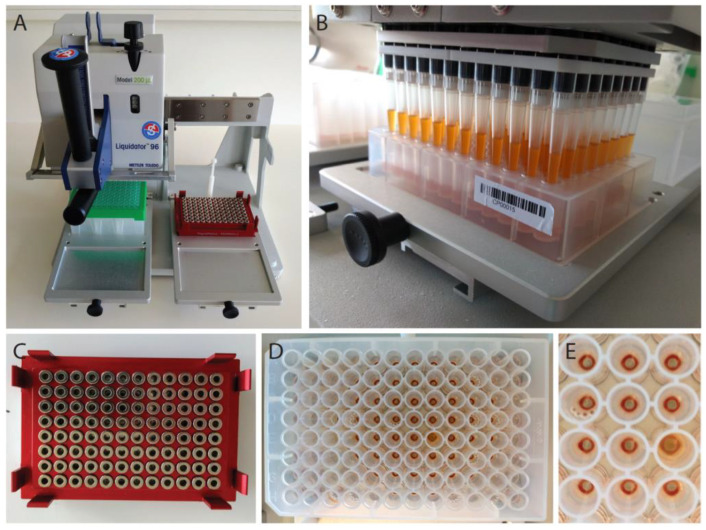
Magnetic bead RNA extraction using the Liquidator 96 pipetting system. (**A**) Liquidator 96, Model 200 µL, tip box is placed in the rear left position and magnet plate (red) in the rear right position. (**B**) The liquidator is used to resuspend magnetic beads in the 96-deep-well plate. (**C**) Magnet plate used to separate the magnetic beads and the supernatant. (**D**,**E**) Ring-shaped pellets are formed after placing the 96-deep-well plate onto the magnetic ring plate (**C**) for 10 min.

**Figure 2 viruses-12-00863-f002:**
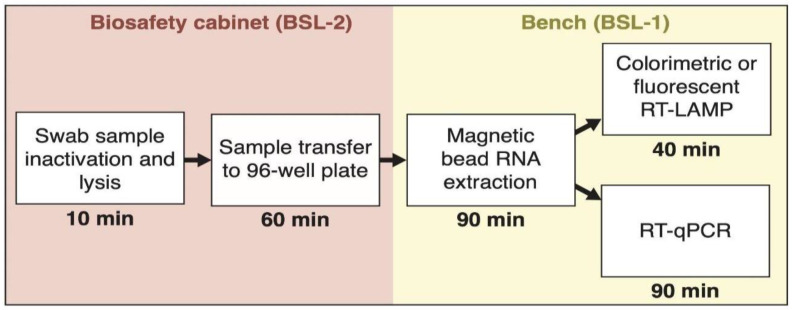
Overview and timeline of sample processing using a magnetic bead RNA purification protocol for SARS-CoV-2 diagnostics.

**Figure 3 viruses-12-00863-f003:**
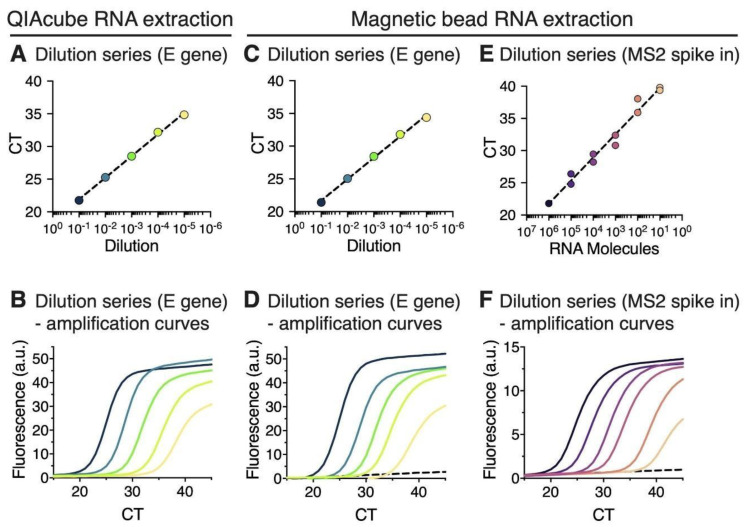
RT-qPCR of RNA extracted from sample dilution series using either the QIAcube RNA extraction kit or the magnetic beads RNA extraction. (**A**–**D**) One positive patient sample was diluted in a 10-fold dilution series from 10^1^ to 10^5^ fold (dark blue—green—yellow). (**A**) RT-qPCR cycle threshold (CT) values of RNA extracted using the QIAcube RNA extraction protocol shown on a semi-logarithmic scale. Semilog non-linear regression shows a goodness of fit of R^2^ = 0.998. (**B**) RT-qPCR amplification curves of RNA extracts shown in (**A**); a.u., arbitrary units. (**C**) RT-qPCR CT values of RNA extracted using the magnetic bead RNA extraction protocol shown on a semi-logarithmic scale. Semilog non-linear regression shows a goodness of fit of R^2^ = 0.996. (**D**) RT-qPCR amplification curves of RNA extracts shown in (**C**). Dashed black line shows an undiluted RNA sample from a SARS-CoV-2 negative patient. (**E**,**F**) Analysis of linearity and detection sensitivity of the magnetic bead RNA extraction protocol using MS2 RNA spike-in. Dilution series of MS2 RNA from 10^1^ to 10^6^ (black—purple—orange), was added into SARS-CoV-2 positive patient samples prior to magnetic bead RNA extraction. (E) CT values for each dilution (duplicates) were plotted against calculated molecule numbers of MS2 RNA per RT-qPCR reaction on a semi-logarithmic scale. All data points and the semilog non-linear regression with a goodness of fit of R^2^ = 0.980 are shown. (**F**) RT-qPCR amplification curves of diluted MS2 RNA extracts shown in (**E**) for one duplicate. Dashed black line is a sample without MS2 spike-in.

**Figure 4 viruses-12-00863-f004:**
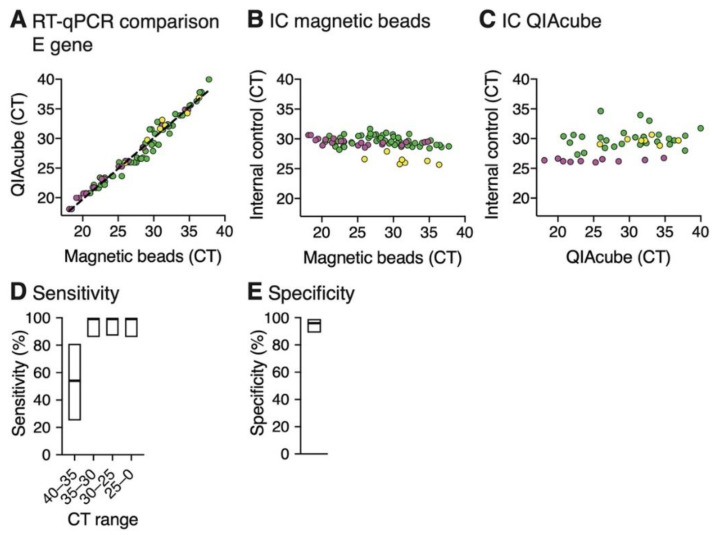
RT-qPCR of RNA extracted from SARS-CoV-2 positive patient samples using magnetic bead RNA extraction and QIAcube RNA extraction. RNA extraction was performed on three sets of SARS-CoV-2 positive patient samples with a total sample number of 82 (yellow = 7 samples, magenta = 20 samples, green = 55 samples) in parallel with either the magnetic bead RNA extraction protocol or the automated QIAcube extraction kit and further analyzed by RT-qPCR using the E gene. (**A**) Comparison of RT-qPCR analysis of RNA extracted by the two different extraction methods. The plot shows the E gene CT values for each sample. A linear regression (equation: f(x) = 1.037x − 1.013, R^2^ = 0.967) is plotted as a dashed line. (**B**,**C**) Analysis of the internal control of the RT-qPCR for RNA extracted by the magnetic bead RNA extraction protocol (**B**) or QIAcube extraction kit (**C**). Plots show the CT values of the internal control plotted against the E gene CT values. (**D**,**E**) Sensitivity (**D**) and specificity (**E**) analyzed by RT-qPCR of the three independent magnetic bead RNA extractions. To determine sensitivity and specificity, QIAcube RT-qPCR results were used as a reference. RT-qPCR after magnetic bead RNA extraction provides close to 100% sensitivity and specificity at a cutoff CT of 35. Data are shown as mean with indicated 95% Clopper–Pearson confidence intervals (Appendix A).

**Figure 5 viruses-12-00863-f005:**
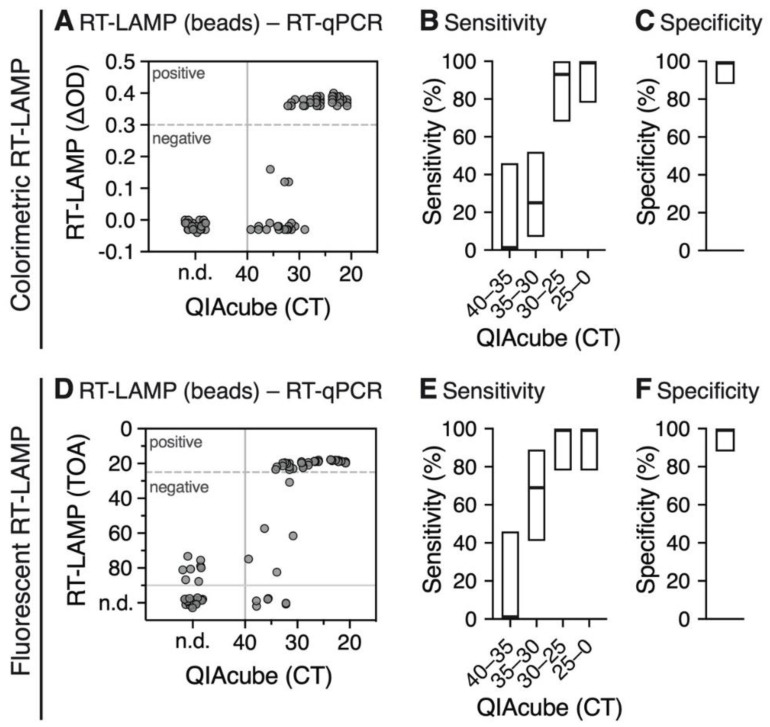
Analysis of colorimetric (**A**–**C**) and fluorescent (**D**–**F**) RT-LAMP of RNA extracted using the magnetic bead RNA extraction protocol compared to RT-qPCR CT values obtained from RNA isolated by the QIAcube extraction. (**A**) Scatter plot shows optical density differences obtained from colorimetric RT-LAMP (ΔOD = OD_434nm−_OD_560nm_) at a time point of 30 min. The ΔOD threshold of 0.3 is indicated as a dashed line. All samples with a CT > 40 were considered as negative (solid line); n.d., not determined. (**B**,**C**) Sensitivity (**B**) and specificity (**C**) of colorimetric RT-LAMP for different CT ranges. The boxes indicate the 95% Clopper-Pearson confidence interval (Appendix A). (**D**) Scatter plot shows time of amplification (TOA) in min for each sample obtained from fluorescent RT-LAMP compared to RT-qPCR CT values obtained from RNA isolated by the QIAcube extraction. The TOA threshold of 25 min (dashed line) was used to define positive and negative samples from RT-LAMP. All samples with a CT > 40 and TOA > 25 min were considered as true negative (solid lines). (**E**,**F**) Sensitivity (**E**) and specificity (**F**) of fluorescent RT-LAMP for different CT ranges. The boxes indicate the 95% Clopper-Pearson confidence interval (Appendix A).

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
