# Peer review of "SARS-CoV-2 RNA Extraction Using Magnetic Beads for Rapid Large-Scale Testing by RT-qPCR and RT-LAMP"

_viruses, 2020, doi:10.3390/v12080863_

Round 1

Reviewer 1 Report

I have reviewed the manuscript entitled “SARS-CoV-2 RNA extraction using magnetic beads
for rapid large-scale testing by RT-qPCR and RT-LAMP” submitted to your esteemed journal by Klein et al.

Proper RNA extraction method is a critical necessity for SARS-CoV-2 virus detection. This manuscript describes a RNA extraction method from SARS-CoV-2 virus using magnetic beads. Although, magnetic beads-based RNA isolation methodology is not novel, the method presented in this manuscript is very timely as it can potentially be applied as an alternative to commercial kits in the clinical laboratories. The effectiveness of this RNA extraction procedure was validated by RT-qPCR and LAMP methods, which was comparable to the commercial QIAcube-based viral RNA extraction method. This manuscript is well presented. I have some comments that the authors may consider for improving the clarity of this manuscript.

General comments:

Although I have little doubt regarding the effectiveness of beads-based RNA extraction method, it is not free from deficiency. One such weakness is the inadvertent loss of RNA during washing before elution. The viral load can be variable in different swabs. Thus, unintentional RNA loss during bead washing may sometimes lead to “false negative” results. In this context, what precautions one should take to avoid improper data interpretation? I suggest the authors to discuss a bit about  potential difficulties  in order to caution the researchers who intends to adapt this method.

Specific comments:

  1. Lines 45: Please mention the full gene name; N gene - nucleocapsid protein, E gene - envelope protein gene, etc

  2. Lines 81-85: Please make it clear how many samples (i.e. swab) were originally collected form COVID patients and from healthy persons. This information should be linked to Lines 212-213 showing 88 positive and 76 negative samples.
  3. The legend to Fig.1 is not clear. The Fig 1A also has other equipment besides Liquidator 96. Possibly using an arrow, or removing other equipment or might be showing a magnified view of Liquidator 96 will be easy to follow.
  4. Lines 129-131: Please clearly state that you studied “envelope protein gene (E gene)”. Also, mention the amplicon size and who deigned primers-probe for RT-qPCR. I assume it is done by Tib-Molbiol.

  5. Lines 138-140: Please mention whether raw CT or normalized CT was used as “input” for statistical analyses.
  6. Lines 176-177: Is it SARS-CoV-2 diagnosis or detection?
  7. Section 2.5: Please provide the RT-qPCR cycling program for MS2 RNA amplification. If it is same as in section 2.4, mention it.
  8. Section 2.6, Line 153: Reference 7 is not appropriate here. In this study, Zhao et al 2020 discusses about the LAMP method, not the LAMP primers targeting the N gene. LAMP primer design is very tricky. Please provide LAMP primer sequences and provide the software used for these primer designs. Or, provide the reference from where these primer sequences were acquired.  
  9. Lines 231-232: “Rates of true and false positives in the RT-qPCR after magnetic beads RNA extraction were calculated accordingly”. Please clarify how the calculation was done. The clarity of Supplementary Tables 1-3 legends largely depends on this statement.

Taken together, this paper has adopted knowledge from the published literature to develop a viable and seemly effective alternative method that can be used for SARS-CoV-2 viral RNA isolation. I have no hesitation in recommending this manuscript for publication after minor revision.

Author Response

Thank you very much for your constructive and valuable comments!

Proper RNA extraction method is a critical necessity for SARS-CoV-2 virus detection. This manuscript describes a RNA extraction method from SARS-CoV-2 virus using magnetic beads. Although, magnetic beads-based RNA isolation methodology is not novel, the method presented in this manuscript is very timely as it can potentially be applied as an alternative to commercial kits in the clinical laboratories. The effectiveness of this RNA extraction procedure was validated by RT-qPCR and LAMP methods, which was comparable to the commercial QIAcube-based viral RNA extraction method. This manuscript is well presented. I have some comments that the authors may consider for improving the clarity of this manuscript.

Author response: Thank you very much for all valuable comments.

General comments:

Although I have little doubt regarding the effectiveness of beads-based RNA extraction method, it is not free from deficiency. One such weakness is the inadvertent loss of RNA during washing before elution. The viral load can be variable in different swabs. Thus, unintentional RNA loss during bead washing may sometimes lead to “false negative” results. In this context, what precautions one should take to avoid improper data interpretation? I suggest the authors to discuss a bit about potential difficulties in order to caution the researchers who intend to adapt this method.

Author response: Indeed, the magnetic bead washing step before the elution could result in the loss of material. However, the RT-qPCR performed on the MS2 and EAV spike-in control showed similar CT values after both magnetic bead and QIAcube extraction (see Figure 4B and 4C), therefore such a brief wash is not sufficient to elute the adsorbed RNA on the beads. Technical loss of RNA during washing in specific samples would be indicated by a change in CT value of the internal control (IC). We added in lines 242–246 “These results also illustrate that quickly washing the beads with water to remove residual ethanol (which could inhibit PCR) does not lead to a substantial loss of RNA. To determine false negatives due to RNA loss and to judge the quality of the RNA extraction, SD of CT values for IC should be calculated. The SD of the CT values for IC should be smaller than 3 CT values and samples with IC CT values with 2 x SD lower IC CT values than the average IC CT should be repeated.

Specific comments:

Lines 45: Please mention the full gene name; N gene - nucleocapsid protein, E gene - envelope protein gene, etc

Author response: We added full gene names in the sentence line 45.

Lines 81-85: Please make it clear how many samples (i.e. swab) were originally collected form COVID patients and from healthy persons. This information should be linked to Lines 212-213 showing 88 positive and 76 negative samples.

Author response: Thank you for this important comment. We clarified the number of patients (lines 82––87): A total of 77 swab samples were collected from 17 SARS-CoV-2 positive and 60 negative patients, which were used for SARS-CoV-2 testing by RT-qPCR. The swab samples were either used at the day of collection (2 positive, 20 negative) or were frozen, stored at -20 °C and thawed just before the RNA extraction (15 positive, 40 negative). In order to facilitate replicate testing, positive and negative samples were further diluted and used in replica to generate 88 positive and 76 negative samples.

In addition, as suggested, we provided the information again in the lines 229––232:

To evaluate the magnetic bead RNA extraction protocol on larger sample sets, 88 SARS-CoV-2 positive and 76 negative samples were generated from 17 SARS-CoV-2 positive patients and from 60 persons negative for SARS-CoV-2, respectively. The samples were subjected to three independent magnetic bead RNA extractions as well as to QIAcube extractions.

The legend to Fig.1 is not clear. The Fig 1A also has other equipment besides Liquidator 96. Possibly using an arrow, or removing other equipment or might be showing a magnified view of Liquidator 96 will be easy to follow.

Author response: Thank you for this comment. We took a new photo of the Liquidator without any other instruments in the background and we have also updated the figure legend.

Lines 129-131: Please clearly state that you studied “envelope protein gene (E gene)”. Also, mention the amplicon size and who designed primers-probe for RT-qPCR. I assume it is done by Tib-Molbiol.

Author response: We clarified the use of the E gene (line 134), mentioned that the primer-probe set was provided by Tib-Molbiol (line 140) and added the amplicon information (line 142-143)

Lines 138-140: Please mention whether raw CT or normalized CT was used as “input” for statistical analyses.

Author response: The CT values were not normalized to the IC control. We added the respective comment in (line 150).

Lines 176-177: Is it SARS-CoV-2 diagnosis or detection?

Author response: It is a workflow that is used to detect the RNA in the swab samples and subsequently conclude a diagnosis. For clarity, we changed the word ‘diagnosis’ to ‘detection’ (line 190).

Section 2.5: Please provide the RT-qPCR cycling program for MS2 RNA amplification. If it is same as in section 2.4, mention it.

Author response: The cycling program for MS2 RNA amplification is the same as in section 2.4. To make this clear we added a sentence (lines 160––161): RT-qPCR cycling program for MS2 RNA amplification was performed like in section 2.4.

Section 2.6, Line 153: Reference 7 is not appropriate here. In this study, Zhao et al 2020 discusses about the LAMP method, not the LAMP primers targeting the N gene. LAMP primer design is very tricky. Please provide LAMP primer sequences and provide the software used for these primer designs. Or, provide the reference from where these primer sequences were acquired.

Author response: Thank you for pointing out this mistake. We have now corrected the reference to Zhang et al, 2020 and added the following sentence with the reference to the primer set sequences:

(Lines 164-166): The RT-LAMP primer set used in this study were targeted against SARS-CoV-2 N gene (Zhang et al., 2020). The sequences and the concentrations of all oligonucleotide in the 10x primer mix used for the RT-LAMP assays can be found in our recent publication (Dao Thi et al, 2020).

Lines 231-232: “Rates of true and false positives in the RT-qPCR after magnetic beads RNA extraction were calculated accordingly”. Please clarify how the calculation was done. The clarity of Supplementary Tables 1-3 legends largely depends on this statement.

Author response: Thank you very much for this comment which helps to improve the clarity of our analysis method. We updated the paragraph on how false and true positives were calculated:

(Lines 249-260): Samples were distributed in small groups of positive and negative samples on the 96-well plate. After magnetic beads RNA extraction and RT-qPCR or RT-LAMP analysis the results were compared to the results of the RT-qPCR after QIAcube extraction to determine true and false positives (TP and FP) and true and false negatives (TN and FN). For the magnetic beads extracted samples a CT cutoff of 40 was used to determine if a sample is defined as positive. The sensitivity was calculated (TP/(TP+FN)) for subsets of samples in specific CT ranges (0––25, 25––30, 30––35, 35––40) (Figure 4D). For all subsets up to CT 35, no false negatives were detected, which leads to a sensitivity of 100% with a 95 % confidence interval (CI) of 86 – 100%. In the range between CT 35––40, six false negatives were counted leading to a sensitivity of 54% (CI = 25 – 81%). These results indicate that false negatives were samples with high CT values and the negative results might be due to fluctuation of the viral load around the limit of detection. In total, three false positives were detected resulting in a specificity of 96% (CI = 89 - 99%) (Figure 4E).

Taken together, this paper has adopted knowledge from the published literature to develop a viable and seemly effective alternative method that can be used for SARS-CoV-2 viral RNA isolation. I have no hesitation in recommending this manuscript for publication after minor revision.

Author response: We want to thank the reviewer again for the detailed and constructive review which helped to improve this manuscript.

Reviewer 2 Report

Klein and authors present a methods paper on extraction and detection of SARS-CoV-2. They demonstrate that, in a background of global reagent shortages for extraction and detection of this pathogen, a diverse range of alternative (more abundant) products can be used to successfully extract viral RNA, and quantify it.

A major strength is the range of methods compared in this study.The authors also do well to show how their methods differ and are similar to approaches such as the Crick study.

My key concern is the way in which the sensitivity of the RT-LAMP assays are presented. The authors need to be much clearer that RT LAMP is considerably less sensitive for the detection of low-abundance/high Ct samples - ie it will produce more false negatives that qPCR. This may be especially important for screening pre-symptomatic cases, where the RNA load is just beginning to build, or for determining whether long-term shedders (eg the elderly) are now truly negative for the virus and can be released from hospital/quarantine. Claiming a high sensitivity only for Cts <30 obscures the significant limitations of RT LAMP detection.

In the methods section, the description of the method for the fluorescent RT LAMP assay and the colorimetric RT LAMP assay are confusing. It needs to be made really clear (probably with subsectioning) that this is TWO different RT LAMP assays and which is which!

Author Response

Thank you very much for your constructive and valuable comments!

Klein and authors present a methods paper on extraction and detection of SARS-CoV-2. They demonstrate that, in a background of global reagent shortages for extraction and detection of this pathogen, a diverse range of alternative (more abundant) products can be used to successfully extract viral RNA, and quantify it. A major strength is the range of methods compared in this study.The authors also do well to show how their methods differ and are similar to approaches such as the Crick study.

Author response: Thank you very much for your positive and constructive review.

My key concern is the way in which the sensitivity of the RT-LAMP assays are presented. The authors need to be much clearer that RT LAMP is considerably less sensitive for the detection of low-abundance/high Ct samples - ie it will produce more false negatives that qPCR. This may be especially important for screening pre-symptomatic cases, where the RNA load is just beginning to build, or for determining whether long-term shedders (eg the elderly) are now truly negative for the virus and can be released from hospital/quarantine. Claiming a high sensitivity only for Cts <30 obscures the significant limitations of RT LAMP detection.

Author response: Thank you very much for this important comment. The RT-LAMP assay is less sensitive than RT-qPCR which can be seen by comparing Fig. 4D and Fig.5B,E. With RT-LAMP samples with a CT of up to 30 can be detected reliably, whereas with RT-qPCR a detection of samples with up to CT 35 is reliable. This was discussed in our previous study by Dao Thi et al. We agree with the reviewer that it is important to also discuss it in the context of this manuscript, so we added the following paragraph:

(Lines 367-377) As we have reported recently (Dao Thi et al, 2020), RT-LAMP to detect SARS-CoV-2 RNA has a decreased sensitivity compared to RT-qPCR: RT-LAMP with the primer set against N gene as used in this study, failed to detect most of the samples with a CT value over 30 measured in a RT-qPCR using the E-Sarbeco primers. Efforts are underway to increase the sensitivity of the RT-LAMP assay, but this is out of the scope of this work. Here, we showed that the presented magnetic bead extraction is compatible with RT-LAMP and does not lower its sensitivity when compared to a QIAcube purification method performed in our previous study (Dao Thi et al, 2020). Either RT-qPCR or RT-LAMP can be used as an independent detection method and both methods do not have to be run in parallel. Due to higher sensitivity, we recommend using RT-qPCR as a detection method after magnetic bead RNA extraction. If resources are limited or large-scale testing is needed, RT-LAMP offers an alternative detection approach, however, its lower sensitivity must be considered for now.

In the methods section, the description of the method for the fluorescent RT LAMP assay and the colorimetric RT LAMP assay are confusing. It needs to be made really clear (probably with subsectioning) that this is TWO different RT LAMP assays and which is which!

Author response: We separated the fluorescence and colorimetric RT LAMP in the materials and methods section into two section 2.6 and 2.7 to make it more clear that these are two different assays.

Reviewer 3 Report

The manuscript by Klein is a well-written and informative report on the use of magnetic beads for extraction of SARS-CoV-2 RNA for rapid large-scale testing by real-time RT-PCR and RT-LAMP.

The information from this manuscript will be of immense value for high throughput molecular testing of SARS-CoV-2 in diagnostic samples during the current, and foreseeable, climate when commercial reagent kits are scarce and expensive.

I only had one comment/suggestion for the authors to consider which related to the throughput of the PCR and RNA-LAMP tests. It may be informative for the reader to have an idea of daily/weekly testing throughput and whether the PCR and LAMP protocols would continue to run together or a separate tests.

Author Response

Thank you very much for your constructive and valuable comments!

The manuscript by Klein is a well-written and informative report on the use of magnetic beads for extraction of SARS-CoV-2 RNA for rapid large-scale testing by real-time RT-PCR and RT-LAMP. The information from this manuscript will be of immense value for high throughput molecular testing of SARS-CoV-2 in diagnostic samples during the current, and foreseeable, climate when commercial reagent kits are scarce and expensive.

Author response: We want to thank the reviewer for this very positive comment.

I only had one comment/suggestion for the authors to consider which related to the throughput of the PCR and RNA-LAMP tests. It may be informative for the reader to have an idea of daily/weekly testing throughput and whether the PCR and LAMP protocols would continue to run together or a separate tests.

Author response: Using the manual pipetting system Liquidator 96, RNA extraction of 96 samples can be performed in 90 minutes, and one person could perform several extractions during a working day of 8 hours. In order to achieve higher throughput, we designed the extraction method using standard 96-well plates so that it will only take minimum effort to adapt it to fully automated liquid handling robotic systems. We used both detection methods in parallel to assure that they both are compatible with magnetic bead extraction, however, it is sufficient to run either RT-qPCR or RT-LAMP as detection method with the consideration that currently RT-qPCR offers better sensitivity. 

We added a sentence regarding the throughput (Line 352-353) and we added a sentence to comment on the RT-qPCR and RT-LAMP that they can be run as independent detection methods (Lines 373-377).

Either RT-qPCR or RT-LAMP can be used as an independent detection method and both methods do not have to be run in parallel. Due to higher sensitivity, we recommend using RT-qPCR as a detection method after magnetic bead RNA extraction. If resources are limited or large-scale testing is needed, RT-LAMP offers an alternative detection approach, however, its lower sensitivity must be considered for now.

Reviewer 4 Report

The concept of using magnetic beads to assist in RNA/DNA isolation is not particularly novel, however the authors do a good job on their approach, data analysis, and commentary. Given the issues with SARS-CoV-2 testing globally, including limitations in RNA isolations supplies, alternatives are needed. Indeed, this reviewer applauds their development of an assay that does not have proprietary components.  Additional points in the discussion, such as the effect of freeze/thawing on CT values, are of value as well.

My only comment is that additional method details, such as length of washes and volume, etc should be in main text and not supplemental.

Author Response

Thank you very much for your constructive and valuable comments!

The concept of using magnetic beads to assist in RNA/DNA isolation is not particularly novel, however the authors do a good job on their approach, data analysis, and commentary. Given the issues with SARS-CoV-2 testing globally, including limitations in RNA isolations supplies, alternatives are needed. Indeed, this reviewer applauds their development of an assay that does not have proprietary components. Additional points in the discussion, such as the effect of freeze/thawing on CT values, are of value as well.

Author response: Thank you very much for this positive summary. We thank you especially for your appreciation of our approach to use as little as possible proprietary components.

My only comment is that additional method details, such as length of washes and volume, etc should be in main text and not supplemental.

Author response: Thank you very much for this comment. We agree that more details for the washing and elution steps during the RNA extraction method should be added to the main text. We updated section 2.2 accordingly:

(Lines 110-119) For each washing step, the 96-well plate was removed from the magnet, resuspended (10×) using a Liquidator 96 until pellets were completely resuspended. The plate was placed back on the magnet for 1 min until the beads formed visible rings. After three washing steps, the magnetic beads were briefly rinsed with 60 µl RNase-free water to remove any residual ethanol while the plate was kept on the magnet. The 96-well plate was visually inspected to ensure that none of the pellets was removed. Finally, nucleic acids adsorbed onto the surface of the magnetic beads were eluted: 50 µl RNAse-free water was added to each well, the 96-well plate was removed from the magnet, resuspended (10×) and vortexed for 5 – 10 min. The 96-well plate was placed back to the magnet until rings of magnetic beads were formed and 50 µl eluate was transferred to a new 96-well PCR plate.
